# Implementation of the Synergy Tool: A Potential Intervention to Relieve Health Care Worker Burnout

**DOI:** 10.3390/ijerph20010489

**Published:** 2022-12-28

**Authors:** Farinaz Havaei, Maura MacPhee, Andy Ma, Vivien W. Wong, Cecilia Li, Irene Cheung, Lina Scigliano, Amera Taylor

**Affiliations:** 1School of Nursing, University of British Columbia, Vancouver, BC V6T 2B5, Canada; 2Fraser Health Authority, New Westminster, BC V3L 3W7, Canada

**Keywords:** Synergy tool, burnout, staff outcomes, staffing, healthcare, workload management, control, participatory action research, quality improvement, collaboration, staff engagement

## Abstract

(1) Background: Healthcare workers experienced rising burnout rates during and after the COVID-19 pandemic. A practice-academic collaboration between health services researchers and the surgical services program of a Canadian tertiary-care urban hospital was used to develop, implement and evaluate a potential burnout intervention, the Synergy tool. (2) Methods: Using participatory action research methods, this project involved four key phases: (I) an environmental scan and a baseline survey assessment, (II), a workshop, (III) Synergy tool implementation and (IV) a staffing plan workshop. A follow-up survey to evaluate the impact of Synergy tool use on healthcare worker burnout will be completed in 2023. (3) Results: A baseline survey assessment indicated high to severe levels of personal and work-related burnout prior to project initiation. During the project phases, there was high staff engagement with Synergy tool use to create patient care needs profiles and staffing recommendations. (4) Conclusions: As in previous research with the Synergy tool, this patient needs assessment approach is an efficient and effective way to engage direct care providers in identifying and scoring acuity and dependency needs for their specific patient populations. The Synergy tool approach to assessing patient needs holds promise as a means to engage direct care providers and to give them greater control over their practice—potentially serving as a buffer against burnout.

## 1. Introduction

The COVID-19 pandemic has had severe consequences for the health and safety of healthcare workers and those under their care. One of these costly consequences is burnout, an occupational syndrome resulting in an overall feeling of disengagement from work [1]. Healthcare worker burnout has been linked to undesirable patient (e.g., adverse events) [1], provider (e.g., pain and suffering) [2], and organizational (e.g., absenteeism, turnover) outcomes [3]. Burnout results from a mismatch between employees’ values and their working conditions, particularly work overload and lack of control [1]. Given the high cost of burnout, healthcare organizations are interested in potential interventions to prevent and/or mitigate the experience of burnout among their health human resources. While healthcare workers were identified as a high-risk population for burnout before the pandemic, there were rising burnout rates during and after the COVID-19 pandemic [4,5,6]. This quality improvement (QI) project was initiated as a practice-academic collaboration between health services researchers and the surgical services program of a tertiary-care urban hospital in British Columbia, Canada. This multi-phased QI project includes the development, implementation and evaluation of a potential burnout intervention that is patient-centered and co-produced with interdisciplinary healthcare teams from four surgical areas. This paper will describe the first two phases, development and implementation, of this burnout intervention QI project. 

### A Patient-Centered Approach to Burnout Intervention

Nurses comprise the greatest proportion of direct patient care providers on acute care units, and nurses’ workload management is significantly associated with nurse, patient and organizational outcomes. A well-documented adverse outcome from unmanageable nursing workloads is burnout [7]. For this QI project, a decision was made by the academic-practice partnership to address burnout by improving nursing workload management—with prospective positive outcomes for other team members and for patients. Nursing workload, in particular, is significantly influenced by patient acuity and dependency needs [7]. The academic-practice partnership, therefore, chose to address burnout by first establishing the patient needs and nursing workload demands for the four surgical areas including Vascular, General, Neuro and Orthopedic Surgery. The partners selected a valid and reliable patient needs assessment tool, the Synergy tool, to objectively determine the patient needs of four surgical patient populations. The Synergy tool is comprised of five patient acuity characteristics (i.e., stability, complexity, predictability, vulnerability, resiliency) and three patient dependency characteristics (i.e., participation in care, participation in decision making and resource availability) that direct care providers score on a three or five-point scale from low to high needs [8,9]. Patient needs scores (i.e., synergy scores) can be used in real-time or aggregated and averaged over time to inform patient care delivery decisions, including decisions regarding staffing levels and skill mix [8]. Given that healthcare staffing decisions are commonly informed by arbitrary standards and/or availability of human resources, patient needs data, generated in real-time with the Synergy tool, holds promise as a burnout intervention that can be efficiently and effectively adapted to different patient contexts.

The Synergy tool has been used predominantly by direct care nurses after conducting their patient assessments, however, recent team-based care studies with nurses and other direct care providers have shown improved direct care providers’ perceptions of workload management, work engagement and control [9,10]. A recent systematic review of 26 international studies, further demonstrates how the Synergy tool is associated with positive care delivery experiences and outcomes for patients and healthcare providers [11]. Despite this positive evidence, the Synergy tool has not been specifically used as a burnout intervention. 

We propose using the Synergy tool to mitigate burnout among healthcare workers because initial evidence suggests that it has potential to facilitate workload management and to foster a sense of control among healthcare workers. According to burnout experts, chronic exposure to work overload contributes to burnout by depleting workers’ capacity to meet their job demands; there is little opportunity to “rest, recover and restore balance” [12]. Likewise, when workers are excluded from decision making in relation to their work, they feel a sense of lack of control and autonomy which subsequently increases their risk of developing burnout and disengaging from their work [12]. 

The Synergy tool facilitates workload management in two important ways (a) using individual patient acuity and dependency needs to inform staffing decisions both in terms of staffing levels and skill mix and (b) using these patient needs to assign staff based on their competencies and capacity to meet patient needs. The former facilitates workload management through ensuring safe staffing and the latter through creating a fit between patient needs and staff competencies [8,9,10]. 

The Synergy tool also fosters healthcare workers’ sense of control by acknowledging the importance of their patient assessment knowledge and skills [13]. This QI initiative requires active involvement of healthcare workers in training workshops that are used for the purpose of adapting and validating the tool as well as co-developing an implementation plan and staffing guidelines. Healthcare workers, therefore, are the ones using the tool to assess and score their patient needs; scores are then used by management and leadership to inform decisions regarding patient care provision and resource distribution. 

## 2. Materials and Methods

Our academic-practice partnership used a participatory action research (PAR) approach to promote inclusive engagement of key stakeholders, including organizational leadership teams and direct care providers through all phases of our burnout intervention QI project [14]. PAR methods can be used in formal research or in QI initiatives to enhance participants’ sense of control, value, and connection with the project [14]. Our QI project consists of four key phases: (I) an environmental scan and a baseline staff survey assessment, (II), a workshop, (III) Synergy tool implementation and (IV) a staffing plan workshop. A follow-up survey to evaluate the impact of Synergy tool use on healthcare worker burnout will be completed in 2023. This QI project was reviewed by university and health authority ethics and considered exempt from research ethics approval, given the partners’ primary focus on work environment improvement via burnout mitigation. 

### 2.1. Phase I

Phase I involved an environmental scan to gain an understanding of the surgical areas’ characteristics, specifically bed and staff size and care team composition. The baseline survey assessment estimated the rate of burnout among staff before the implementation phase. All direct care nursing and allied health staff and front-line leadership from the four surgical areas were invited to complete an electronic survey designed by the researchers and distributed by the leadership team. 

The anonymous survey included demographics questions and questions on staff workplace experiences including burnout, workload management, involvement and influence. Burnout was measured using the validated 19-item Copenhagen Burnout Inventory (CBI). The CBI covers three domains (Personal burnout, 6 items; Work-related burnout, 7 items; Client-related burnout, 6 items) with response options ranging on a five-point Likert-type scale from 0 lowest severity (To a very low degree/Never) to 100 highest severity (To a very high degree/Always) [15]. Mean scores were obtained for each type of burnout: personal burnout refers to the degree of physical and psychological exhaustion experienced by the person; work-related burnout is the degree of exhaustion due to their work; and client-related burnout is the degree of exhaustion due to work with clients/patients. Burnout scores per surgical area were categorized as low (<50), moderate (50–74), high (75–99) and severe (100) [16].

The pre-implementation survey also included two subscales on workload management and involvement/ influence from the validated Guarding Minds at Work Survey [17]. Each subscale consists of five items asking participants to indicate their level of agreement or disagreement on a four-point scale ranging from strongly disagree (1) to strongly agree (4). For each subscale, sum scores of 5–9 indicate serious concern, 10–13 significant concern, 14–16 minimum concern, and 17–20 relative strength [17]. Survey responses were analyzed using descriptive statistics. 

### 2.2. Phase II

Phase II involved a 2-day workshop for project teams from the four surgical areas. Each implementation unit focused on an overview of the Synergy tool, tool adaption and validation for their specific patient populations and co-development of an implementation plan. Project teams included direct care providers from nursing, allied health and front-line managers. Allied health team members included occupational therapy (OT), physiotherapy (PT), dietary and social work (SW) team members for every participating unit except general surgery (See Table 1 for units’ team composition). After familiarizing participants with the Synergy tool, teams worked together to identify specific assessment indicators for high, moderate and low acuity and dependency needs for their respective patient populations (Appendix A show Adapted Synergy Tools for each Surgical Area). Each team then refined their assessment indicators through scoring practice with case vignettes representing a diverse range of their patients. Patient vignettes were also used to determine inter-rater reliability, which ranged from 90% to 95% across the respective workshops for each of the four areas. The latter part of the workshop was used for participants to establish an implementation plan for using the Synergy tool with site-specific assessment indicators to score their patient populations. 

### 2.3. Phase III

During Phase III, project team members scored all patients across morning and night shifts between 14 to 16 times during the following data collection periods: Vascular (26 April–31 May 2022), General (27 March–12 May 2022), Neuro (1 January–2 February 2022) and Orthopedics (14 March–7 April 2022). A three-point scale was used to indicate low, moderate and high acuity and dependency needs per patient. Scoring was done by nurses and allied health providers who participated in the workshop. The teams used hard copy data collection: Patient-level Synergy scores were later transferred to an Excel document for the researchers to analyze. 

Synergy scores were analyzed using descriptive statistics to gain an understanding of the patient profile on each Surgical area. To visualize the changes in patient acuity and dependency needs over time, subscale mean scores were created per patient assessment and also averaged for each day and each surgical area. 

### 2.4. Phase IV

Phase IV was a half-day workshop for project teams to share, discuss, and validate Phase III results. The workshop goals were to answer the following question: “How should your surgical area be staffed in terms of staff levels and skill mix for different levels of care needs and under various patient profiles during the day and night shifts?” During the workshop, participants used the data from Phase III to create staffing guidelines for typical patients with high, moderate and low acuity and dependency needs. Examples of typical patients are presented as case vignettes in Appendix A. 

## 3. Results

Table 1 shows the results of the environmental scan describing the characteristics of each surgical area. Each area has approximately 30 regular and 4 overflow beds. Average patient length of stay is between six (General) to 10 (Vascular) days. A total of 251 interdisciplinary team members work across the surgical program with approximately one regulated nurse (registered nurse [RN] and licensed practical nurse [LPN]) caring for 3.5–4.25 patients on the day shift and 1 to 4.25–5 patients on the night shift. The Neuro and Orthopedic areas use an RN/LPN skill mix as opposed to the Vascular unit with a RN/LPN/care aide (CA) skill mix and the General unit with RN/CA skill mix. 

Table 2 provides the overall and area-specific characteristics and experiences of survey respondents. The survey response rate from 251 staff members was 73% (n = 182). On average, survey respondents were 36 years old with 8 years of experience. An overwhelming majority of participants were female (96%), direct care providers (95%), RN (80%) and held a fulltime position (64%). In terms of burnout, personal and work-related burnout were most concerning with 49% and 47% of respondents across all areas qualifying as high to severe personal and work-related burnout, respectively. A much smaller proportion of 12% of the respondents met the criteria for patient-related burnout. In relation to workload management, between 83% (Orthopedics) to 90% (General) of respondents across all four areas had serious or significant concerns about this aspect of their work environment. In relation to involvement and influence, there was a lot more diversity across the four areas from 24% (Orthopedics) to 55% (General). 

Table 3 shows demographics information of patients and their Synergy scores. Overall, a total of 96 (Vascular) to 131 (General) patients were scored in each of the surgical areas with every patient being scored between 2.9 to 4.2 times on average. The Vascular area (M = 70.1, SD = 11.4) had the oldest patient population as opposed to the General Surgical area (M = 59.3, SD = 17.9) that had the youngest. There was a relatively even distribution of male and female patients in Neuro and Orthopedic areas as opposed to Vascular and General Surgical areas that had a greater proportion of male (~60%) versus female patients. 

Table 3 also provides descriptive statistics on Synergy scores that demonstrate each area’s patient needs profile. For example, Vascular patients were highest needs in relation to complexity (M = 2.11, SD = 1.03) and participation in care (M = 2.67, SD = 1.30); General Surgical patients were highest needs in relation to complexity (M = 2.73, SD = 1.32) and stability (M = 3.17, SD = 1.07); Neurosurgical patients were highest needs in relation to predictability (M = 1.86, SD = 1.00) and vulnerability (M = 2.12, SD = 1.14); and Orthopedics patients were highest needs in relation to complexity (M = 2.79, SD = 1.38) and participation in care (M = 2.13, SD = 1.17). Participation in decision making was identified as the lowest needs patient characteristic across all four areas with mean scores ranging between 4.37 to 4.69. Subscale mean scores showed that Neurosurgical patients had the highest level of acuity (M = 2.33, SD = 0.78) and dependency (M = 3.09, SD = 0.85) needs.

Figure 1 demonstrates average acuity and dependency scores for each unit over different periods of time. For example, Figure 1 suggests an increase in both acuity and dependency needs for Surgical patients in early April that partially reverses later in the month; and comparing units, the averaged scores indicate greater variability for Vascular patients’ needs, relative to other units. 

Table 4 presents the nurse staffing recommendations developed by the workshop participants based on each Surgical area’s Synergy scores. Overall, the interdisciplinary project teams agreed that Synergy scores provided an accurate depiction of the patient profile on their corresponding surgical area. Each surgical area noted and recommended a staffing increase of at least one nurse designation for the day and the night shifts. The greatest staffing gap belonged to the Neuro and Orthopedic Surgical areas. In Neuro Surgery, workshop participants agreed that there was a gap of one RN, one LPN and two CAs for the day shift and two RNs and one LPN for the night shift. In Orthopedic Surgery, Synergy scores showed a gap of one RN and two CAs for the day and two RNs and two CAs for the night shifts. The staffing discussions also resulted in recommended changes in relation to allied health staffing not presented here.

## 4. Discussion

The Synergy tool has been used extensively for care delivery redesign based on patient needs assessment data and staffing complements based on Synergy scores. These care design projects conducted by academic researchers [9,18,19], have been particularly effective when project planning, implementation and evaluation have involved key stakeholder groups, particularly direct care providers, their management and patient representatives [10,20,21,22]. 

Within one Canadian province, two urban emergency departments (ED) used the Synergy tool and Synergy scores to address staff burnout and staff perceptions of the quality-of-care delivery [10]. A PAR approach, similar to this study, was used to engage two interdisciplinary ED teams in one large Canadian city. The teams used the Synergy tool to document the need for more full-time nurses and for additional social services to assist service users with social care/dependency needs. Validated survey tools were used pre-post Synergy tool implementation, and one important finding was a significant decrease in “crisis mode activity,” indicating providers’ capacity to slow down and engage more fully in their work. The researchers concluded that more time to communicate and to integrate team-based care delivery may buffer against crisis mode workload associated with burnout [10]. A qualitative component to this ED study found that staff became more focused on their own and their patients’ relational needs [22]. 

Although research evidence identifies positive staff outcomes associated with the Synergy tool, to our knowledge, the tool has not been specifically used as a burnout mitigation strategy. The theoretical rationale underlying our approach is based on high involvement work practices (HIWPs), which are evidence-based best practices to increase employee engagement, job satisfaction and retention in their work settings [23]. A well-known HIWP model is known as PIRK, where P represents power or employee empowerment, (I) is increased access to information, (R) are work-related rewards and (K) are knowledge and skills acquisition, such as professional development opportunities [23]. HIWPs are ‘bundles’ of evidence-based practices associated with improved employee outcomes, including decreased burnout. HIWPs are also being tied to the Institute for Healthcare Improvement Quadruple Aim of designing work environments that improve the staff experience with subsequent positive impacts on patient care, including the cost and quality of care delivery [24]. 

Kilroy et al. used validated survey tools to test relationships between the PIRK model HIWPs, job demands and burnout. These researchers showed that employee access and engagement in HIWPS did not increase work-related demands (a common concern); instead, employees reported decreased job demands and decreased burnout when HIWP opportunities were offered to them [23]. The researchers concluded that HIWPs give autonomy and control back to employees so that they can better manage their work demands and consequently avoid burnout from emotional exhaustion, withdrawal and depersonalization [23]. 

Meacham et al. conducted a multi-level survey study to determine the buffering effects of HIWPs at different systems levels including organizational (policies, resources), supervisory (relational support) and individual (resilience strategies) on nurse work engagement and intention to leave [25]. This study found that when HIWP organizational, relational and individual resources are available to nurses, they have significantly more positive attitudes about their work and their intention to stay [25]. The researchers recommended that healthcare organizations provide multiple sources of HIWPs to build up employees’ relational reserves at the local level of work [25]. 

This QI project adds to evidence that the Synergy tool may be a multi-pronged approach to mitigating burnout: (a) by addressing work environment factors associated with improved provider outcomes, such as control over practice and workload management; (b) by acknowledging and valuing providers’ expertise at patient assessment, care planning and delivery; and (c) by generating rigorous assessment-driven data about patient care needs to inform staffing decisions and resource allocation. Our high staff engagement in project participation indicates the importance of professional autonomy and engagement to our practice partners.

### Study Limitations

Limitations of this QI project include hard copy data collection: Leadership is planning to integrate Synergy tool assessment and scores within the electronic health record. Another limitation of this project is lack of follow-up evaluation data at this point in time. If this project is successful, we hope to apply for research funding to carry out systematic integration and evaluation of Synergy tool use throughout the organization. For example, an ideal would be to conduct a randomized controlled study to ensure comparable work environments, patient populations and staffing complements before and after a Synergy tool intervention. 

## 5. Conclusions

We propose adding the Synergy tool to other evidence-based HIWPs, given our initial successes at engaging direct healthcare providers and management in a patient-centered, solution-focused approach to workload management concerns. This QI project with Synergy tool use acknowledges that direct care staff are the experts when it comes to ‘fit’ between staff competencies and patients’ priority care needs. Hopefully our planned post-implementation survey in 2023 will establish the link between Synergy tool use as a potential HIWP and decreased potential for burnout [24,25]. 

## Figures and Tables

**Figure 1 ijerph-20-00489-f001:**
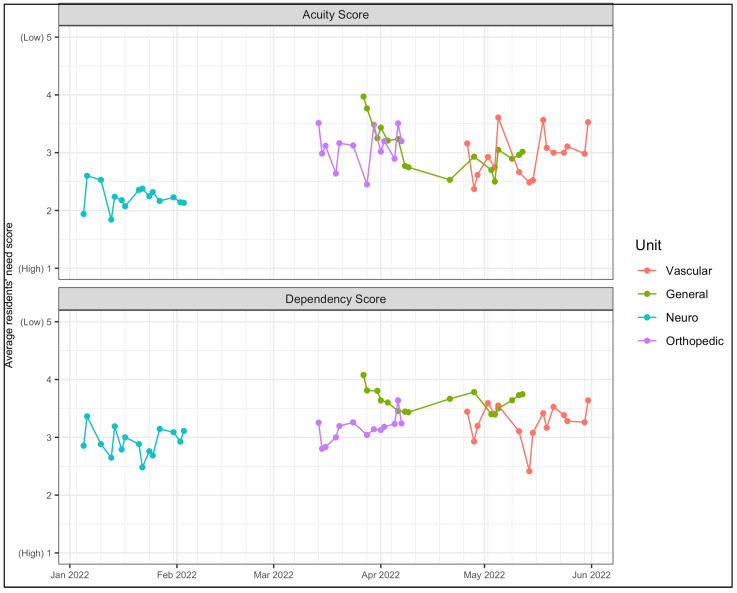
Trends Overtime in Patient Acuity and Dependency Scores across each Unit.

**Table 1 ijerph-20-00489-t001:** Unit Characteristics Based on the Environmental Scan.

Surgical Specialty	Vascular	General	Neuro	Ortho
Bed size (n)	30	30	31	30
Overflow bed size (n)	4	4	4	4
Length of stay (M days)	10	6	7–8	8
Staff size (n)	55	55	67	74
Day shift nurse staffing	7 RNs 2 LPNs 1 CA	8 RNs 0 LPN 1 CA	9 RNs 1 LPN 0 CA	7 RNs 2 LPNs 0 CA
Night shift nurse staffing	7 RNs 1 LPN 0 CA	7 RNs 0 LPNs 0 CA	8 RNs 1 LPN 0 CA	6 RNs 2 LPNs 0 CA
Years of nursing experience	1–2	1–2	15	1–2
Care team composition	Surgeon, RN, LPN, CA, NP, Vascular Wound Care nurse, CNE, PCC, OT PT, SW Dietitian, Pharmacist	Surgeon, Residents, Hospitalists, RN, CA, NP, Enterostomal Wound Care Nurse, OT, PT, Pharmacist	Surgeons, Hospitalist, Residents RN, LPN, NP, UC, PCC, Charge Nurse, CNE, PT, OT, RA, SW, SLP, RT, Dietitian, Pharmacist	Surgeons, Hospitalists, Resident, RN, LPN, NP UC, PCC, Charge Nurse, CNE, PT, OT, RA, SW, SLP, RT, Dietitian, Pharmacist

Note: RN, registered nurse; LPN, licensed practical nurse; CA, care aide; NP, nurse practitioner; OT, occupational therapist; PT, physiotherapist; UC, unit clerk; PCC, patient care coordinator; CNE, clinical nurse educator; RA, rehabilitation assistant; SW, social work; SLP, speech language pathologist; RT, respiratory therapist.

**Table 2 ijerph-20-00489-t002:** Overall and Area-Specific Survey Respondent Characteristics and Experiences.

	Vascular	General	Neuro	Ortho	Total
Total surveys distributed (n)	55	55	67	74	251
Valid survey responses (n)	42	39	42	59	182
Response rate (%)	76%	71%	63%	80%	73%
Age [M (SD)]	37.1 (10.7)	35.9 (8.9)	39.1 (10.1)	33.1 (9.9)	35.9 (10.1)
Gender (%)					
Male	2.5	2.6	2.4	5.3	3.4
Female	97.5	97.4	97.6	93.0	96.0
Prefer to describe	0.0	0.0	0.0	1.8	0.6
Experience years [M (SD)]	8.7 (7.4)	8.0 (6.6)	11.0 (7.7)	5.8 (6.4)	8.1 (7.2)
Role (%)					
Direct care provider	95.1	94.9	95.2	94.8	95.0
Leadership/management	2.4	2.6	2.4	1.7	2.2
Educator	0.0	0.0	0.0	0.0	0.0
Support/Ancillary Staff	0.0	2.6	2.4	3.4	2.2
Other	2.4	0.0	0.0	0.0	0.6
Designation (%)					
LPN	14.3	0.0	2.4	16.9	9.3
RN	78.6	87.2	83.3	72.9	79.7
RPN	0.0	0.0	0.0	0.0	0.0
CA	0.0	2.6	2.4	0.0	1.1
Allied Health (e.g., OT, PT, SW)	4.8	7.7	11.9	10.2	8.8
Physician	0.0	0.0	0.0	0.0	0.0
Other, please describe	2.4	2.6	0.0	0.0	1.1
Employment status (%)					
Full-time	64.3	56.4	73.8	62.7	64.3
Part-time	21.4	33.3	11.9	13.6	19.2
Casual	14.3	10.3	14.3	23.7	16.5
Personal burnout [M, (SD)]	73.4 (18.2)	72.9 (24.3)	62.1 (20.6)	69.5 (21.3)	69.4 (21.4)
No/Low (%)	10	24	31	17	20
Moderate (%)	36	11	38	36	31
High (%)	48	55	26	39	41
Severe (%)	7	11	5	8	8
Work-related burnout [M, (SD)]	69.9 (21.2)	74.6 (19.6)	61.6 (22.6)	68.5 (21)	68.6 (21.4)
No/Low (%)	12	15	26	14	16
Moderate (%)	31	26	43	42	36
High (%)	55	59	29	37	44
Severe (%)	2	0.0	2	7	3
Patient-related burnout [M, (SD)]	40.5 (22)	42.3 (25)	35.1 (19)	46.8 (25.4)	41.6 (23.4)
No/Low (%)	64	56	71	55	61
Moderate (%)	24	31	26	27	27
High (%)	12	10	2	12	10
Severe (%)	0	3	0	5	2
Workload management [M, (SD)]	10.6 (2.8)	10.2 (3.1)	10.3 (3.3)	11.2 (2.7)	10.6 (3)
Serious concern (%)	38.1	46.2	35.7	22	34.1
Significant concern (%)	47.6	43.6	45.2	61	50.5
Minimum concern (%)	9.5	5.1	16.7	11.9	11.0
Relative strength (%)	4.8	5.1	2.4	5.1	4.4
Involvement & Influence [M, (SD)]	14.7 (2.5)	13.2 (2.8)	13.1 (3.3)	14.1 (2.5)	13.8 (2.8)
Serious concern (%)	4.8	13.2	17.1	8.6	10.6
Significant concern (%)	21.4	42.1	29.3	15.5	25.7
Minimum concern (%)	54.8	31.6	43.9	65.5	50.8
Relative strength (%)	19	13.2	9.8	10.3	12.8

Note: RN, registered nurse; LPN, licensed practical nurse; RPN, registered psychiatric nurse; CA, care aide.

**Table 3 ijerph-20-00489-t003:** Demographics of patients and their Synergy ratings.

Surgical Specialty	Vascular	General	Neuro	Orthopedic
N	96	131	111	110
Scoring frequency per patient [Mean (SD)]	4.2 (3.9)	2.9 (2.8)	3.2 (2.5)	2.9 (2.0)
Age [Mean (SD)]	70.1 (11.4)	59.3 (17.9)	62.3 (17.4)	64 (22.3)
Gender (%)				
Male	60.4	61.8	52.3	49.1
Female	39.6	38.2	47.7	50.9
Stability [Mean (SD)]	2.85 (1.27)	3.17 (1.07)	2.69 (1.09)	3.07 (1.03)
Low (%)	24.0	26.7	13.5	23.6
Moderate (%)	45.8	58.8	56.8	60.0
High (%)	30.2	14.5	29.7	16.4
Complexity [Mean (SD)]	2.11 (1.03)	2.73 (1.32)	2.39 (0.99)	2.79 (1.38)
Low (%)	4.2	21.4	6.3	29.1
Moderate (%)	39.6	45.8	53.2	34.5
High (%)	56.2	32.8	40.5	36.4
Predictability [Mean (SD)]	3.69 (1.11)	3.54 (1.03)	1.86 (1.00)	3.32 (1.05)
Low (%)	42.7	36.6	3.6	31.8
Moderate (%)	51.0	56.5	31.5	57.3
High (%)	6.2	6.9	64.9	10.9
Resiliency [Mean (SD)]	3.55 (1.07)	3.48 (1.16)	2.59 (0.94)	3.78 (1.29)
Low (%)	41.7	38.9	7.2	61.8
Moderate (%)	49.0	51.9	61.3	21.8
High (%)	9.4	9.2	31.5	16.4
Vulnerability [Mean (SD)]	3.37 (1.00)	3.58 (1.18)	2.12 (1.14)	3.28 (1.40)
Low (%)	30.2	42.0	7.2	38.2
Moderate (%)	59.4	47.3	38.7	38.2
High (%)	10.4	10.7	54.1	23.6
Participation in DM [Mean (SD)]	4.62 (0.73)	4.61 (0.80)	4.37 (0.75)	4.69 (0.62)
Low (%)	88.5	86.3	73.0	90.0
Moderate (%)	9.4	11.5	27.0	9.1
High (%)	2.1	2.3	0	0.9
Participation in care [Mean (SD)]	2.67 (1.30)	3.65 (1.29)	2.48 (1.32)	2.13 (1.17)
Low (%)	19.8	51.1	15.3	10.0
Moderate (%)	37.5	36.6	42.3	36.4
High (%)	42.7	12.2	42.3	53.6
Resource availability [Mean (SD)]	2.96 (0.69)	3.25 (0.78)	2.43 (1.09)	2.96 (0.84)
Low (%)	7.3	18.3	10.8	11.8
Moderate (%)	82.3	75.6	48.6	73.6
High (%)	10.4	6.1	40.5	14.5
Acuity score	3.11 (0.72)	3.30 (0.87)	2.33 (0.78)	3.25 (0.84)
Low (%)	11.5	22.9	5.4	22.7
Moderate (%)	83.3	71.0	60.4	69.1
High (%)	5.2	6.1	34.2	8.2
Dependency score	3.42 (0.69)	3.84 (0.73)	3.09 (0.85)	3.26 (0.60)
Low (%)	20.8	46.6	14.4	12.7
Moderate (%)	75.0	50.4	73.0	84.5
High (%)	4.2	3.1	12.6	2.7

Note: DM, decision making.

**Table 4 ijerph-20-00489-t004:** Nurse staffing recommendations informed by Synergy scores.

Surgical Specialty	Vascular	General	Neuro	Orthopedic
Day shift nurse staffing recommendation	+1 RN	+1 RN +1 CA	+1 RN +1 LPN +2 CA	+1 RN +2 CA
Night shift nurse staffing recommendation	+1 CA	+2 LPNs	+2 RN +1 LPN	+2 RN +2 CA

Note: +, increase; RN, registered nurse; LPN, licensed practical nurse; CA, care aide.

## Data Availability

The data presented in this study are available on request from the corresponding author.

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
