# Peer review of "Implementation of the Synergy Tool: A Potential Intervention to Relieve Health Care Worker Burnout"

_ijerph, 2022, doi:10.3390/ijerph20010489_

Round 1
Reviewer 1 Report
Havaei et al implemented the Synergy tool in 4 surgical wards to study the mitigational effect on burnout.
The idea behind the intervention is that the Synergy tool allows assessment of patient needs and subsequently might empower nursing staff to meet the demands of their patients in a more controlled and less burnout-inducing manner.
The manuscript describes the implementation carefully and precisely, I have no comments whatsoever, but it does not describe the effect of the intervention (the implementation of the Synergy tool) on the incidence or prevalence of burnout, and this is my only but major concern. The authors acknowledge this and announce that the results will be reported at a later stage, but still, it feels somewhat preliminary to publish this now. Hence they should refrain from claiming that the intervention has shown to mitigate burnout at this stage, as they do now in for instance the conclusion of the abstract.
Apart from this I loudly applause the authors efforts, congratulate them on accomplishing this difficult but important study and I would encourage acceptance and publication.
Author Response
Dear Reviewers,
Many thanks for your thoughtful review of our paper and your recommendations for edits. We have added your revision requests and our responses below.
Reviewer 1: Hence they should refrain from claiming that the intervention has shown to mitigate burnout at this stage, as they do now in for instance the conclusion of the abstract.
Response: We added the word “potentially” to the abstract Conclusion. We also changed the language throughout the manuscript to reflect potential versus actual.
On lines 81-83 we modified the sentences to say: “We propose using the Synergy tool to mitigate burnout among healthcare workers because initial evidence suggests that it has potential to facilitate workload management and to foster a sense of control among healthcare workers.”
On line 303, we modified our statement: “This QI project adds to evidence that the Synergy tool may be a multi-pronged approach to mitigating burnout:
Reviewer 2 Report
Thank you for the opportunity to review the submitted article titled "Implementation of the Synergy Tool: A potential intervention to relieve health care worker burnout".
The authors present an innovative approach to the phenomenon of burnout by assessing the demand for patient care as a means of direct involvement of direct care providers, which, by controlling the tasks performed, can counteract professional burnout as well as serve to help assess minimum staffing standards. The article presents the preliminary results of the project, which is interesting. However, I have some important considerations and concerns.
There are doubts about the inclusion in the study of surgical departments with different care needs, e.g., vascular surgery and neurosurgery, where neurosurgery, as is well known, requires more staffing, as obviously demonstrated in the study. It would be reasonable to compare the needs of patients residing in departments with similar standards and workloads, as this has a significant impact on the level of burnout.
190-193 – please present a list of the team of providers participating in the study.
In 291-295, the authors claim that: "This QI project adds to evidence that the Synergy tool is a multi-pronged approach 291 to mitigating nurse burnout". My question is whether all direct care groups were assessed using the Copenhagen Burnout Inventory or just nurses? This should be clarified in the introduction, as the participation of such different groups of providers in the study may result in a "blurring" of the results of the burnout scale. At the same time, the lack of opportunity to review the Synergy tool may also affect the above doubts.
In the study, the authors included information about the abandonment of the ethical procedure due to the qualification of the study as a quality improvement. This is reasonable for assessing the needs of patients, but how to qualify the Copenhagen Burnout Inventory (CBI) questionnaire for assessing burnout and addressing many aspects of respondents' psychosocial lives in the above study?
In 260-263, the authors write: "to our knowledge, the tool has not been specifically used as a burnout mitigation strategy". During the various phases of the study presented in this paper, how did the authors conclude that levels of burnout would be reduced when, as written in lines 17 and 103: "A follow-up survey to evaluate the impact of Synergy tool use on healthcare worker burnout will be completed in 2023"? Wouldn't it make sense at this stage of the preparation of the publication to use only data from the Synergy tool with regard to care demand and staffing planning without correlation with job burnout? An assessment of the respondents' level of burnout to correlate with the preliminary results conducted before the implementation phase and after the project is completed in 2023 will be able to determine whether or not the level of burnout will be reduced in this case of implementing changes to the workstation based on the results obtained with the Synergy tool.
To sum up: the range of needs reported by our patients has an impact on the workload of providers and the workplace is an important aspect of business management. However, this is only one of many aspects that can affect job burnout.
Author Response
Reviewer 2: It would be reasonable to compare the needs of patients residing in departments with similar standards and workloads, as this has a significant impact on the level of burnout.
Response: We have added the importance of a more controlled research study in our Limitations.
Lines 316-319: “ For example, an ideal would be to conduct a randomized controlled study to ensure comparable work environments, patient populations and staffing complements before and after a Synergy tool intervention. “
Reviewer 2: 190-193 – please present a list of the team of providers participating in the study.
Response: Rather than a table, we added clarification to lines 143-148: “Project teams included direct care providers from nursing, allied health and front-line managers. Allied health team members included occupational therapy (OT), physiotherapy (PT), dietary and social work team members for every participating unit except general surgery (See Table 2 for units’ team composition).”
Reviewer 2: In 291-295, the authors claim that: "This QI project adds to evidence that the Synergy tool is a multi-pronged approach 291 to mitigating nurse burnout". My question is whether all direct care groups were assessed using the Copenhagen Burnout Inventory or just nurses? This should be clarified in the introduction, as the participation of such different groups of providers in the study may result in a "blurring" of the results of the burnout scale. At the same time, the lack of opportunity to review the Synergy tool may also affect the above doubts.
Response: We added clarifying sentences throughout the manuscript. We also felt it might be helpful to state a bit more about why we used a tool associated with nurses’ workload management. The Synergy Tool was initially intended for nurse use only, but the PAR process we describe in this paper has been extended in recent studies with the Synergy Tool to engage all key patient care stakeholders.
Added lines are 50-55:
“Nurses comprise the greatest proportion of direct patient care providers on acute care units, and nurses’ workload management is significantly associated with nurse, patient and organizational outcomes. A well-documented adverse outcome from unmanageable nursing workloads is burnout [7]. For this QI project, a decision was made by the academic-practice partnership to address burnout by improving nursing workload management—with prospective positive outcomes for other team members and for patients.”
We modified lines 72-75: “The Synergy tool has been used predominantly by direct care nurses after conducting their patient assessments, however, recent team-based care studies with nurses and other direct care providers have shown improved direct care providers’ perceptions of workload management, work engagement and control [9,10].
On lines 81-83 we modified the sentences to say: “We propose using the Synergy tool to mitigate burnout among healthcare workers because initial evidence suggests that it has potential to facilitate workload management and to foster a sense of control among healthcare workers.”
On lines 238-241: “Overall, the interdisciplinary project teams agreed that Synergy scores provided an accurate depiction of the patient profile on their corresponding surgical area. Each surgical area noted and recommended a staffing need increase of at least one nurse designation for the day and the night shifts.”
On line 119 we clarified who participated in the baseline survey: “All direct care nursing and allied health staff and front-line leadership from the four surgical areas were invited to complete an electronic survey designed by the researchers and distributed by the leadership team.”
On lines 303-310: “This QI project adds to evidence that the Synergy tool may be a multi-pronged approach to mitigating burnout: a) by addressing work environment factors associated with improved provider outcomes, such as control over practice and workload management; b) by acknowledging and valuing direct care providers’ expertise at patient assessment, care planning and delivery ; and c) by generating rigorous assessment-driven data about patient care needs to inform staffing decisions and resource allocation. Our high staff engagement in project participation indicates the importance of professional autonomy and engagement to our practice partners.”
Reviewer 2: In the study, the authors included information about the abandonment of the ethical procedure due to the qualification of the study as a quality improvement. This is reasonable for assessing the needs of patients, but how to qualify the Copenhagen Burnout Inventory (CBI) questionnaire for assessing burnout and addressing many aspects of respondents' psychosocial lives in the above study?
Response: In this QI project, burnout is an important outcomes indicator for evaluating the merits of the Synergy Tool as a burnout intervention. We decided to use the CBI to gather burnout information for all key stakeholders on the four units per the request of our practice partners. Other data we used to create unit profiles at the request of our practice partners. All data from individuals are anonymous: We will not be using these data for research purposes.
Reviewer 2: In 260-263, the authors write: "to our knowledge, the tool has not been specifically used as a burnout mitigation strategy". During the various phases of the study presented in this paper, how did the authors conclude that levels of burnout would be reduced when, as written in lines 17 and 103: "A follow-up survey to evaluate the impact of Synergy tool use on healthcare worker burnout will be completed in 2023"? Wouldn't it make sense at this stage of the preparation of the publication to use only data from the Synergy tool with regard to care demand and staffing planning without correlation with job burnout?
Response: The baseline assessment of provider burnout in this QI project is helpful background for establishing burnout types and levels for key patient care providers, nurses, allied health and managers. We felt these baseline data would add to readers’ appreciation of the burnout dilemma in healthcare settings.
Reviewer 2: To sum up: the range of needs reported by our patients has an impact on the workload of providers and the workplace is an important aspect of business management. However, this is only one of many aspects that can affect job burnout.
Response: There are many factors that affect job burnout, and we decided to focus on workload management, given the significant positive relationship between burnout and nursing workload. Some team-based studies with the Synergy tool have now shown that the tool may be a burnout buffer for other providers as well—by improving their workload management and sense of professional autonomy.
We added lines 50-55 to hopefully explain the evolution of Synergy Tool use from nursing patient needs assessment tool to a potential healthcare team burnout intervention. “Nurses comprise the greatest proportion of direct patient care providers on acute care units, and nurses’ workload management is significantly associated with nurse, patient and organizational outcomes. A well-documented adverse outcome from unmanageable nursing workloads is burnout [7]. For this QI project, a decision was made by the academic-practice partnership to address burnout by improving nursing workload management—with prospective positive outcomes for other team members and for patients.”